# Non-Coding RNAs and Innate Immune Responses in Cancer

**DOI:** 10.3390/biomedicines12092072

**Published:** 2024-09-11

**Authors:** Carlos Romero Díaz, María Teresa Hernández-Huerta, Laura Pérez-Campos Mayoral, Miriam Emily Avendaño Villegas, Edgar Zenteno, Margarito Martínez Cruz, Eduardo Pérez-Campos Mayoral, María del Socorro Pina Canseco, Gabriel Mayoral Andrade, Manuel Ángeles Castellanos, José Manuel Matías Salvador, Eli Cruz Parada, Alexis Martínez Barras, Jaydi Nora Cruz Fernández, Daniel Scott-Algara, Eduardo Pérez-Campos

**Affiliations:** 1Tecnológico Nacional de México/IT Oaxaca, Oaxaca 68030, Mexico; carlos.romero@itoaxaca.edu.mx (C.R.D.); e_mily_3@hotmail.com (M.E.A.V.); mcruz@itoaxaca.edu.mx (M.M.C.); eli.cruz.parada@gmail.com (E.C.P.); 2Consejo Nacional de Humanidades, Ciencias y Tecnologías (CONAHCyT), Facultad de Medicina y Cirugía, Universidad Autónoma “Benito Juárez” de Oaxaca (UABJO), Oaxaca 68020, Mexico; mthernandez@conahcyt.mx; 3Centro de Investigación, Facultad de Medicina UNAM-UABJO, Universidad Autónoma “Benito Juárez” de Oaxaca (UABJO), Oaxaca 68020, Mexico; lperezcampos.fmc@uabjo.mx (L.P.-C.M.); eperezcampos.fmc@uabjo.mx (E.P.-C.M.); mpina.cat@uabjo.mx (M.d.S.P.C.); gmayoral.fmc@uabjo.mx (G.M.A.); drajaydi@yahoo.com.mx (J.N.C.F.); 4Facultad de Medicina, Universidad Nacional Autónoma de Mexico, Ciudad de México 04510, Mexico; ezenteno@unam.mx (E.Z.); mangeles_castellanos@unam.mx (M.Á.C.); 5Hospital General Dr. Aurelio Valdivieso, Oaxaca 68040, Mexico; matiasjosemanuel8492@hotmail.com; 6Facultad de Ingeniería, Universidad Autónoma de Querétaro, Querétaro 76017, Mexico; saxbarras@outlook.com; 7Unité de Biologie Cellulaire des Lymphocytes and Direction of International Affairs, Institut Pasteur, 75015 Paris, France; 8Laboratorio de Patología Clínica “Dr. Eduardo Pérez Ortega”, Oaxaca 68000, Mexico

**Keywords:** cancer, piRNA, miRNA, siRNA, specificity, immune-inducible, epimutation, transfer, inheritance, trained immunity, chronic inflammation

## Abstract

Non-coding RNAs (ncRNAs) and the innate immune system are closely related, acting as defense mechanisms and regulating gene expression and innate immunity. Both are modulators in the initiation, development and progression of cancer. We aimed to review the major types of ncRNAs, including small interfering RNAs (siRNAs), microRNAs (miRNAs), piwi-interacting RNAs (piRNAs), and long non-coding RNAs (lncRNAs), with a focus on cancer, innate immunity, and inflammation. We found that ncRNAs are closely related to innate immunity, epigenetics, chronic inflammation, and cancer and share properties such as inducibility, specificity, memory, and transfer. These similarities and interrelationships suggest that ncRNAs and modulators of trained immunity, together with the control of chronic inflammation, can be combined to develop novel therapeutic approaches for personalized cancer treatment. In conclusion, the close relationship between ncRNAs, the innate immune system, and inflammation highlights their importance in cancer pathways and their potential as targets for novel therapeutic strategies.

## 1. Introduction

Until 2018, cancer was the second leading cause of death worldwide. The most prevalent types in men were lung, prostate, colorectal, stomach, and liver cancer, while in women they were breast, colorectal, cervical, lung, and thyroid cancers [1]. Cancer can be cataloged as a group of diseases that affect different kinds of organs due to uncontrolled cells growing and eventually spreading to other organs, a process called metastasis, the latter being the major cause of death [2].

Innate immunity is part of a complex system that plays a role in defense against pathogens and, similarly, against cancer cells [3,4]. Different mechanisms participate in this system such as those of the skin, mucous, secretions, phagocytes (i.e., neutrophils, monocytes, macrophages), and inflammation-related proteins, among others. Among the repertoire of the immune system, a new strategy has been discovered: the Trained Innate immunity (TII). It develops when innate immune cells are exposed to stimuli that induce changes in them and their progeny in the bone marrow, thus preparing them to respond quickly and effectively to subsequent challenges with the same stimuli [5,6]. An important role for TII in chronic inflammation and cancer has been suggested. In this context, TII stimulation can eliminate neoplastic cells [7,8].

ncRNAs, like innate immunity, play a role in inflammation and may have deleterious or beneficial effects on cancer through upregulation of the inflammasome and inflammatory pathways. There is considerable evidence that ncRNAs play a critical role in human malignancies [9,10]. ncRNAs are also involved in processes associated with tumorigenesis, such as uncontrolled cell proliferation, apoptosis, genomic stability or instability, epigenetic regulation or deregulation, uncontrolled transcription of oncogenes and tumor suppressor genes, metabolic regulation or deregulation, regulation or deregulation of apoptosis, immune escape, tumor metastasis, O-GlcNAcylation in cancer, and lipid metabolism of tumors [11,12,13]. The regulation of ncRNAs in cancer stem cells involves various signaling pathways, several of which are involved in tumor resistance to antineoplastic therapies.

ncRNAs have been implicated in a wide range of human cancers and their signaling pathways, such as the Wing-less-related integration site/β-catenin (Wnt/β-catenin), Notch and Hedgehog (Hh) signaling pathways in solid tumors [14], the Hippo/Yes-associated protein 1 (Hippo/Yap) signaling pathway and the transforming growth factor β/Suppressor of Mothers against Decapentaplegic (TGF-β/SMAD) signaling in hepatocellular carcinoma [15,16], the Notch and the p53 signaling pathway in breast cancer [17,18,19], the Wnt/*β*-catenin signaling pathway in prostate cancer [20], and Hedgehog/Glioma-associated oncogene homolog 1 Hedgehog-Gli1 signaling in neural changes associated with pancreatic ductal adenocarcinoma [21]. Other pathways that are also altered in other tumors include the Wingless/Integrated (Wnt) signaling pathway in colorectal cancer [22], the phosphatidylinositol 3-kinase/protein kinase B/mammalian target of rapamycin (PI3K/AKT/mTOR) and the mitogen-activated protein kinase (MAPK) signaling pathway in lung cancer [22,23], and the Janus kinase/signal transducers and activators of transcription (JAK/STAT) pathway in cervical cancer [24].

Advances in cancer therapy have been achieved by combining several approaches; the similarities in the mechanism of action of trained immunity and ncRNA could be exploited to improve response to new therapeutic approaches, as discussed below. However, characteristics of the innate immune response and TII such as inducibility, specificity, memory, and transfer have not been compared in detail with ncRNAs.

Although the two systems are different, they share defense mechanisms. It is not known why ncRNA and TII share common mechanisms; however, ncRNA deregulation favors the imbalance of apoptosis, abnormal tumor metabolism, tumor stem cell phenotype, mesenchymal-epithelial transformation, immune checkpoints, cytokine regulation, and tumor exosome content [25].

The objective of this study was to compare the similarities between the innate immune system and ncRNA with a focus on cancer, as well as the combination with chronic inflammation, with the aim of identifying a potential new therapeutic approach. To achieve this, we conducted a focused search and discussed the selection of articles for analysis with a group of co-authors. Searches were conducted in PubMed, Google Scholar, and the Cochrane Library from 1 January 2023, to 29 February 2024, using the search terms: “trained innate immunity”, “trained immunity”, “innate immunity”, “innate immune response”, “pattern recognition receptors”, pattern recognition receptors (PRRs), “inducibility”, “specificity”, “memory”, “specificity”, “memory”, “transfer”, “heterogeneity”, “inducible”, “genetic maintenance”, “non-coding RNAs”, “ncRNAs”, “small non-coding RNAs”, “sncRNAs”, “cancer”, and “tumor”.

At the same time, we cross-referenced these terms with the following: “neutrophils”, “macrophages”, “innate immune cells”, “chronic inflammation”, P-element induced wimpy testis in Drosophila (PIWI) “PIWI-interacting RNA”, PIWI-interacting RNA (piRNA), MicroRNAs (microRNA), MicroRNAs (miRNA), “small interfering RNA”, “siRNA”, “transferable”, “heritable”, “epimutation”, and “transgenerational epigenetic inheritance”.

In addition, we did not consider the terms “medium non-coding RNA” or “mid-size noncoding RNAs” (mncRNAs) as exclusion criteria.

## 2. Main Types of Regulatory Non-Coding RNAs (ncRNAs)

ncRNAs have been arbitrarily classified according to their genomic position (intragenic, intergenic) and their shape (circular, linear) [26]. ncRNAs are also divided into two types according to their function: housekeeping and regulatory. Housekeeping ncRNAs include ribosomal RNAs (rRNAs), transfer RNAs (tRNAs), small nuclear RNAs, and small nucleolar RNAs. Regulatory non-coding RNAs are classified according to their length. Short ncRNAs (<200 nt) include piwi-interacting RNAs (piRNAs) and microRNAs. Long non-coding RNAs (lncRNAs, ≥500 nt) include linear RNAs (lincRNAs) and circular RNAs (circRNAs) among the most studied [27]. Another classification according to their length includes three types of ncRNAs: long non-coding RNAs (lncRNAs), medium non-coding RNAs (mncRNAs), and small non-coding RNA (sncRNAs) several of which have regulatory functions [28,29,30] (Table 1). ncRNAs can modify genetic information, and some are even involved in placing stable epigenetic marks on DNA that are passed on through cell divisions and generations. There is evidence that ncRNAs interact with each other and that their deregulation is associated with diseases such as cancer [31].

In humans, the proportion of lncRNA genes and sncRNA genes is greater than that of protein-coding genes. Of 62,700 reported genes, 30.93% are protein-coding genes, 31.77% are lncRNA genes, 12.06% are sncRNA genes, and 23.50% are pseudogenes (published on GENCODE, 2022 update; https://www.gencodegenes.org, accessed on 19 June 2024). The Declaration of the Consensus Group has agreed that long non-coding RNAs are regulatory sequences and can be divided into three categories: small RNAs of less than 50 nucleotides (nt), of ~50–500 nt, and of more than 500 nt [67]. The activity of ncRNA depends on its length, biogenesis, subcellular location, and circulation, such as some miRNAs found in mitochondria, endoplasmic reticulum, and RNA granules [68].

The functions of lncRNAs include regulation of chromatin, transcription, interference with signaling pathways, neuronal differentiation, hematopoiesis, and immune response; they are also implicated in cancer initiation and progression [69]. lncRNAs are found in both the nucleus and cytoplasm and can be linear or circular in shape; they are usually multi-exonic and have a high degree of alternative splicing [70].

mncRNAs are very diverse and are involved in the maintenance and integrity of the genome, as well as in the modulation of gene expression in response to environmental signals [71]. They consist of large families and are involved in a wide range of physiological and pathological processes such as cancer [72]. Like sncRNAs, they play a crucial role in the regulation of gene expression.

sncRNAs have a different pathway of biogenesis and a different function in the cell; for example, siRNAs media RNA interference (RNAi) and piRNAs are involved in genome maintenance by transposable elements [73]. sncRNAs are non-coding regulatory sequences between 20 and 50 nt; they are species-specific, present in viruses, plants, and animals, and differ in how they are generated and transported in eukaryotic organisms. There are three major families of sncRNAs in eukaryotic organisms: miRNAs, piRNAs, and siRNAs.

Double-stranded RNA molecules known as siRNAs act in the RNA interference pathway to silence the expression of specific genes [74]. siRNAs are derived from a long double-stranded RNA molecule that is cleaved into fragments of 20 to 25 nt by the Dicer enzyme. These fragments are incorporated into the RNA-induced silencing complex (RISC), which identifies and cleaves mRNA complementary to siRNA [75]. Dicer is involved in siRNA and miRNA biogenesis. Dicer dysregulation may be associated with neurological, psychiatric, autoimmune diseases, and cancer [76]; therefore, endonucleases such as Dicer are involved in genetic maintenance by ncRNAs [77].

miRNAs are nucleotide sequences found in prokaryotes and eukaryotes. In eukaryotes, they are in several different compartments such as the rough endoplasmic reticulum, processing bodies, endosomes, lysosomes, mitochondria, nucleus, and extracellular fluids in vesicles or associated with proteins such as Argonaute-2 [78]. miRNAs regulate the expression of more than 60% of genes, despite representing only 2% to 3% of the human genome [79].

## 3. Mechanisms of Genetic Maintenance and the Innate Immune Response

Several systems maintain homeostasis in living organisms [80], including one for immune response and another for gene maintenance. From a mechanistic point of view, both systems are composed of stimuli (external or internal), sensors or receptors, a control center (which is the detector and controller of errors), effectors (which determine the value of all system variables), and the variables (all the molecules and their indicators that remain within a range of values compatible with life) [81].

Genome integrity is very important since the preservation of the species depends on it. Genome maintenance requires the concerted action of cellular metabolism, the cell cycle, and DNA repair activities, which together constitute the genome maintenance pathways.

Chromosomal instability has been linked to deregulation of the innate immune response [82,83,84] and cancer and involves multiple complex intra- and interchromosomal rearrangements [85] with multiple breakpoints, which have been termed chromothripsis [86]. This dysregulation involves ncRNAs, such as miRNAs, which could be associated with cancer initiation, progression, or metastasis (upregulation of miRNAs or oncomiRs), whereas downregulated miRNAs may act as tumor suppressors [87]. There are numerous examples where deregulation of genetic maintenance tilts the environment in favor of the tumor, e.g., defects in transcriptional enhancers alter gene expression programs and contribute to tumorigenesis [88], deletions of immunostimulatory factors can alter lymphocyte proliferation and lead to inadequate antitumor responses [89], and epigenetic alterations in extracellular signaling domains can facilitate cancer progression by altering the interaction of tumor cells with their environment [90].

The mechanisms of ncRNAs in cancer may involve different molecular pathways such as cell signaling, cell cycle, apoptosis, angiogenesis, invasion, metastasis, and drug resistance. In the latter, ncRNAs can modulate drug efflux, cell apoptosis, and autophagy [91]; for example, there are long non-coding RNAs (lncRNAs) that promote tumors, such as LINC01559 and UNC5B-AS1; these are upregulated in pancreatic ductal adenocarcinoma (PDAC) and are regulators of aerobic glycolysis in PDAC [92], as well as modulators of innate and acquired immunity through the tumor necrosis factor alpha (TNF-α) [93].

ncRNAs are involved in multiple cellular processes including the modulation of innate immunity [94,95], differentiation, and tumor development, and are currently a therapeutic target in clinics [96]. Numerous studies demonstrate their involvement in antiviral defense and immunity against cancer and autoimmune diseases, e.g., miRNAs regulate host defense mechanisms against viruses, bacteria, and fungi, while lncRNAs act as competing RNAs that block miRNAs from binding to mRNA [97,98]. 

The number of ncRNAs associated with cells and regulatory molecules in the immune response of cancer, including miRNA, lncRNA, and circRNA and excluding piRNAs, currently recorded in the RNA2Immune database reaches 4348 immune molecules-ncRNA associations and 485 immune cell-ncRNA associations in 138 cancer types [99]. In this database, liver cancer, gastric breast cancer, non-small cell lung cancer, colorectal cancer, osteosarcoma, glioma, prostate cancer, melanoma, and lung cancer are among the tumors with the most reported ncRNA associations [99].

The relationship between ncRNAs, trained immunity, and cancer has also paved the way for new treatment options such as β-glucan and Bacillus Calmette–Guérin (BCG), both of which can induce local inflammation and native interferon-gamma (IFN-γ) induction. Among the receptors for adoptive immunity are dectin-1, which is the β-glucan receptor, and Nucleotide-binding oligomerization domain-containing protein 2 (NOD2), which is the receptor for the muramyl peptide of *Mycobacterium bovis* in the BCG vaccine [100].

Maintenance mechanisms may also be affected, e.g., dysregulation of both innate immunity and ncRNA is observed with the p50-associated cyclooxygenase-2 extragenic RNA (PACER), which acts as a transcriptional regulator by interacting with the nuclear factor kappa-light-chain-enhancer of activated B cells (NF-κB) pathway. PACER is also involved in the regulation of R-loops, which are three-stranded structures consisting of an RNA-DNA hybrid and the remainder of the translocated DNA strand [101]. PACER is involved in both inflammation and arachidonic acid metabolism and has been implicated in the deregulation of cyclooxygenase-2 (COX-2) in lung cancer [102,103] and dependent trained immunity, specifically against metastatic lung cancer [104].

## 4. Epigenetic, Non-Coding RNAs, Chronic Inflammation and Cancer

Epigenetics or transgenerational epigenetic inheritance (TEI) is the regulation of gene expression without changes in DNA sequence [105,106]. Mechanisms involved in heritable epigenetic changes include DNA methylation, histone modifications, acetylation, phosphorylation, and ubiquitination. ncRNAs are also regulators of the epigenetic status of human DNA [107,108]. Two types of epigenetic inheritance are recognized: ‘intergenerational’ and ‘transgenerational’; the first transmits the change to its immediate descendants, i.e., F1, but then the changes are lost. In the transgenerational type, the epigenetic change must be inherited in the absence of the stimulus for several generations after F2 or F3 [109,110].

Experimental evidence in animals suggests that sncRNAs are molecules capable of changing and transmitting epigenetic information across generations [97]. The relationship between epigenetics, ncRNAs, and cancer could involve the hypermethylation of miRNA promoters, such as miR-34, miR-342, and miR-345, which favors the reduction of tumor suppressor miRNAs and leads to the overexpression of oncogenes as in colorectal cancer [111]. HOX transcript antisense intergenic RNA (HOTAIR) is a lncRNA that is overexpressed in renal cancer and promotes metastasis by recruiting histone modifiers that affect both histone methylation and demethylation to silence tumor suppressor genes [112,113]. Furthermore, epigenetic drugs modify the reading or erasure of epigenetic marks either by acting on methylations or as post-translational modifications, such as histone deacetylase inhibitors (HDACi), e.g., vorinostat, romidepsin, belinostat, and panobinostat can modulate miRNA expression and induce proliferation arrest, angiogenesis, and sensitivity to apoptosis [114].

In cancer, many factors are involved in the tumor inflammatory microenvironment, both tumor-promoting mechanisms and stromal components, as well as those involved in tumor-suppressing immunity (Figure 1).

Although there is a plethora of evidence in different models, the following paragraphs provide a series of examples to illustrate the relationship between innate immunity, chronic inflammation, ncRNAs, and cancer using breast cancer as a model.

The breast cancer microenvironment is highly inflammatory. It consists of a broad spectrum of cells and factors, including infiltrating immune cells (macrophages, dendritic cells, natural killer cells, myeloid-derived suppressor cells, mast cells, and granulocytes), cytokines, and growth factors that coordinately contribute to carcinogenesis and tumor progression. Several miRNAs and lncRNA are involved in the regulation of inflammatory cytokines, cell differentiation, homeostasis, immune checkpoint signaling pathways, apoptosis, necroptosis, cell cycle, cell proliferation, and invasion, i.e., depending on the circumstances, miRNAs can act as a tumor suppressor or as an oncogene.

miR-382 inhibits breast cancer progression and metastasis by affecting the M2 polarization of tumor-associated macrophages [115] but might induce an inflammatory response. miR-21 expression is associated with tumor growth and metastasis [116], by suppressing tropomyosin alpha-1 (TPM1) and programmed cell death 4 (PDCD4), affecting the mammalian target of rapamycin (mTOR) pathway. Also, miR-21 downregulates PTEN, Suppressor of Mothers Against Decapentaplegic (Smad7) [117], Methionine adenosyl transferase II alpha (MAT2A), StAR-related lipid transfer domain protein 13 (STARD13), and Zinc Finger Protein 132 (ZNF132) [118]. In breast cancer, miR-21 has been associated with apoptosis via B-cell lymphoma 2 (BCL2) [117]. Increased serum miR-21 levels in HER2-positive breast cancer patients predict survival in patients receiving neoadjuvant chemotherapy combined with trastuzumab [119].

At the onset of breast cancer, one escape mechanism is the loss of tumor cell antigenicity through increased LINK-A expression, which blocks the presentation of breast cancer antigens [120]. This mechanism has been observed in triple-negative breast cancer (TNBC) cells, as well as in other tumors such as renal papillary cell carcinoma, ovarian serous cystadenocarcinoma, and renal clear cell carcinoma. Loss of antigenicity favors tumor survival and expansion. The use of LINK-A locked nucleic acids (LNA LINK-A) or G-protein–coupled receptor (GPCR) antagonists in mice improves the stability of antigen peptide-loading complex (PLC) and major histocompatibility complex class I (MHC I), thereby improving antigen presentation and the potential use of combined immunotherapy [120]. In addition, the downregulation of LINK-A inhibits cell viability, colony-forming ability, and cell migration in non-small cell lung cancer [121]. LINK-A has also been implicated in inflammatory processes involving IL-1β and CXCL16 through the Link-A/HB-EGF/HIF1α feedback loop, which promotes obesity [122]. Obesity is associated with a low-grade inflammatory microenvironment [123] called “metainflammation”, which is mediated by macrophages [124]. These are regulated by several signaling pathways, such as the JAK/STAT pathway, CCAAT-enhancer-binding proteins (C/EBP), Peroxisome Proliferator-Activated Receptor gamma (PPARγ), Interferon regulatory factors (IRFs), and by lncRNAs (lncRNA E330013P06), as well as by several miRNAs (miR223, miR155, miR125b and Let7c) [125]. Combinations of these miRNAs, such as miR-1246+, miR-206+, miR-24+, and miR-373 have been proposed as markers with very high sensitivity and specificity in the diagnosis of breast cancer [125].

miR-122 is related to both obesity and breast cancer. Studies have shown that miR-122 levels are substantially linked with body mass index (BMI) and weight loss in breast cancer survivors, highlighting its participation in molecular pathways connecting obesity and breast cancer [126,127]. There have also been reports that miR-122 has a strong association with chronic inflammatory processes [128,129].

lcnRNAs may also be regulators of the maturation of important components of innate immunity. Dendritic cell differentiation is controlled at least by STAT3-binding lncRNA-dendritic cells (lnc-DC) in humans [130]; in breast cancer, lnc-DC can inhibit tamoxifen-induced apoptosis by upregulating antiapoptotic (Bcl2 and Bcl-xL). lnc-DC stimulates the production of cytokines, which in turn activates STAT3. Upregulation of lnc-DC is related to a poor prognosis. The lnc-DC expression can predict tamoxifen efficacy and could be used as a predictor of tamoxifen response [131]. In contrast, among the variety of lncRNAs deregulated in breast cancer, there is BC069792. This acts as a tumor suppressor gene and inhibits the proliferation, invasion, and metastasis of breast cancer cells [132].

## 5. Specificity of Non-Coding RNAs and the Immune System

ncRNAs and the immune system are specific and non-specific; there are several similarities and differences between both systems that are discussed below (Table 2). The specificity of sncRNAs depends on complementarity; they are highly complementary and thus specific to RNA sequence. In addition, miRNAs can be partially or almost perfectly complementary; when miRNAs are partially complementary, they inhibit the translation of their complementary mRNAs [133].

Cells involved in trained immunity include NK cells [144], monocytes, macrophages, neutrophils [145], and immune progenitor cells in the bone marrow [6]. These cells are involved in the modulation of hypermutable states, e.g., there is evidence for interferon-dependent innate immunity in the control of preleukemic clones encoding the oncogenic transcription factor ETV6-RUNX1, which can progress to B-cell precursor acute lymphoblastic leukemia (BCP-ALL) [146]. It has been proposed that the pathogenic mechanisms for BCP-ALL include an imbalance between regulators of proliferation, such as long non-coding RNA “colorectal neoplasia differentially expressed” (LncRNA CRNDE), and regulators of apoptosis [147], such as miR-345-5p and abnormal the cytokine response [146,148] (Figure 2). Furthermore, this interrelation between ncRNA and trained immunity leads us to consider that the use of modulators of both systems could prevent or improve the treatment of some cancers, as in the case of BCP-ALL where the application of β-glucan could prevent the development of BCP-ALL [146].

The specificity of siRNAs is limited by several factors such as cross-hybridization, sequence-specific binding to cellular proteins, and non-specific “dsRNA reaction”, among others [136]. While miRNA-mediated silencing occurs by translational repression (partial complementation) or degradation (almost perfect complementation), either deadenylation or exonuclease activity can occur [149]. The target range of piRNAs is determined by PIWI-interacting RNAs, while their specificity is determined by multimeric complexes, such as sleep-like domain proteins with highly specific nucleolytic activities [150].

In the case of innate immunity, the system is based on specificity, e.g., Toll-like receptor (TLR), proteins that recognize specific sequences of RNA, lipopolysaccharides, or peptide sequences present in viruses or bacteria. TLR-3 recognizes viral dsRNA, promotes cytokine expression, and can be pro-tumor or anti-tumor. TLR3 activated by polyinosinic:polycytidylic acid [poly(I:C)] can induce apoptosis and autophagy in melanoma, breast, and ovarian cancer. TLR-4 activated by lipopolysaccharide (LPS) can promote tumor growth and metastasis in colorectal, gastric, and pancreatic cancer [138,151].

miR-29a, miR-29b, and miR-29c facilitate specific activation of NK cell responses by targeting the B7-H3 immune checkpoint in neuroblastoma, opening the possibility of combining treatments using miRNAs and immunotherapy with dinutuximab [152].

## 6. Transgenerational Epigenetic Inheritance

siRNAs, miRNAs, and piRNAs, as well as other sequences such as lncRNAs, participated in epigenetic regulation; although, the most studied in the field of epigenetics are miRNAs. The mechanism of action of these is through inhibitory enzymes that participate such as DNA methyltransferases (DNMTs), histone modifications, and chromatin remodeling [153]. The methylation mechanism is associated with other epigenetic mechanisms, such as histone modification [154]. Methylation frequently occurs at CpG (5′—cytosine—phosphate—guanine—3′). These short stretches of palindromic DNA containing large amounts of CpG dinucleotides (cytosine followed by guanine) are called “CpG islands” [155]; they are found in promoters (regions of DNA where transcription begins). Due to cytosine methylation in CpG, there is a high rate of CpG  >  TpG mutation, which is related to epigenetic inheritance and evolution [156].

In cancer, stable changes in DNA methylation occur, called epimutations; an example of epimutation is the hypermethylation of miR-663a, which can trigger CpG island methylator phenotype high (CIMP-H) endometrial cancer [157]. This aberrant methylation/epimutation of the miR-663a promoter has been observed in normal tissue from endometrial cancer patients [158].

In a study, cross-fostering and mating among nursing siblings were postulated using a/a and Avy/a mice, revealing down- and up-regulation of miR-186-5p and Gsk3b, respectively. This has reportedly been linked to ovarian cancer and may have an impact on epigenetic inheritance that could be transferred across generations [158].

Other examples of epigenetic changes are reported in Colorectal Cancer Metastasis by DNA methylation and histone modification [159]. The presence of mutated histones called oncohistones [160] such as H3 K27M in pediatric brain cancers [161,162] and H3.3 G34R/V in giant cell tumors of bone (GCTB) was observed [163].

## 7. Transfer of Trained Immunity

β-glucan-induced trained immunity is associated with transcriptomic and epigenetic changes in granulopoiesis [164]. These changes lead to the reprogramming of neutrophils towards an anti-tumor phenotype, which is transferable by bone marrow transplantation in naive mice. Trained immunity is a concept associated with the reprogramming of innate immunity cells, whereby when these cells are specifically exposed to molecules such as β-glucan from *Candida albicans*, *Trametes versicolor*, *Saccharomyces cerevisiae*, LPS, BCG, and cholesterol crystals [165] respond much better to a secondary stimulus, i.e., one that does not correspond to a transcriptional program [8]; e.g., pre-metastatic macrophages are reprogrammed with whole β-glucan particles [166,167].

An example of immunostimulation of trained immunity combined with ncRNA is the use of β-1,3-glucan-type structures, such as schizophyllan (SPG), which together with miR-155 and the CpG-ODN adjuvant delayed tumor growth in mice [168].

## 8. Inducibility of ncRNA

Inducibility has been observed in ncRNAs, e.g., in SPHK1, an enzyme that catalyzes the phosphorylation of sphingosine to S1P and ceramide [169]. S1P promotes cell proliferation and survival, while ceramide is involved in cell cycle arrest and apoptosis. SPHK1 plays an important role in non-small cell lung cancer (NSCLC). In NSCLC, miR-495 is decreased and associated with poor prognosis, while miR-495-3p acts on SPHK1 to induce lethal mitophagy and reduce tumor proliferation [170].

Another example of inducibility has been reported for withaferin, which, derived from *Withania somnifera* (Solanaceae), induces cytotoxicity in triple-negative breast cancer (TNBC) cells by upregulating at least ten tumor suppressor microRNAs, including miR-181c-5p. Furthermore, this miR reduces cell proliferation in TNBC cells through withaferin A and miR-181c-5p, mimicking a cotreatment strategy [171]; overexpression of miR-181c-5p increases the macrophage phagocytic capacity [172].

An example of inducibility is the mTOR, a key regulator of cellular metabolism, cell growth, and immune response. It is regulated by several microRNAs, including miR-99a, miR-100, and miR-199a, which directly inhibit its expression or activity [62,173]. Therefore, hyperactivation of mTOR is crucial in the development of invasive breast cancer; an anti-breast cancer effect can be produced by cell cycle arrest that induces mTOR-specific siRNA through apoptosis [174]. On the other hand, miR-122, mentioned above and associated with obesity and breast cancer, and miR-34a are increased after anthracycline treatment. It has been suggested that chemotherapy-induced miRNAs are derived from both the tumor and the non-tumor compartments [175].

There are miRNA networks that regulate the dynamics of immune and cancer cells, such as miR-155, which binds CpG oligodeoxynucleotides (CpG-ODNs), promotes T-cell-mediated tumor immunity, and can induce a more potent immune response in many types of cancer [172,176].

## 9. Heterogeneity of ncRNAs

Like many systems, the innate immune response and ncRNAs are heterogeneous and have specific functions. In the innate immune response, TLRs are heterogeneous in structure, location, ligands, and signaling pathways; in addition, differential expression of TLRs is observed between normal and malignant cells [177].

The heterogeneity of ncRNAs can be observed in terms of location and structure as well as expression; their heterogeneity is modified by various mechanisms, such as alternative splicing and processing of ncRNA transcripts, epigenetic modifications of DNA and histones, as well as editing of ncRNAs and transposable elements [178].

An example of heterogeneity in ncRNAs is reported in a case study of the intertumoral distribution of miRNA-20a and miRNA-125b expression profiles in luminal A or luminal 2 (Her2) subtypes of breast cancer. Significant differences in miRNAs were found, particularly between normal breast, tumor center, contralateral tumor periphery, and tumor margin, which could be associated with poor prognosis [179]. Also, circulating miR-20a is reported in patients with cervical cancer [180]. miRNA-20a is part of the miR-17–92 cluster which includes several miRNAs and has been implicated in the regulation of inflammation and the function of invariant natural killer T (iNKT) cells, also known as classical or type I NKT cells [181,182]. Deregulation of miRNA-20a is associated with inflammatory disorders and infectious diseases [183].

Another example where heterogeneity has been reported is with miR-21, which is upregulated in colorectal, breast, lung, and pancreatic cancer. Furthermore, the expression and function of miR-21 may vary depending on the tumor microenvironment [184]. Likewise, induction of miR-21 appeared to mediate disease progression and metastasis in p53-deficient tumor keratinocytes [185,186].

## 10. Transfer of ncRNAs

The movement or exchange of genetic material is known as genetic transfer. When the transfer occurs between organisms of the same or different species, it is termed horizontal; when it occurs from parents to their offspring, it is called vertical [187]. For example, HPV is the major pathogen associated with cervical cancer and, in this infection, HPV gene transfer requires positively charged HPV L2 and L1 protein sequences through the heparan sulfate receptor [188,189]. In addition, cervical cancer ncRNAs are related to inflammation and are regulated by high mobility group 1 (HMGB1) such as miR-34a, miR-1284, and miR-142 [190].

Transfer is one of the most studied properties of the acquired adaptive immune response, it has also been described in ncRNAs; however, it has been little studied in trained immunity [191].

Microvesicles (MVs) appear to be the most common form of extracellular communication or delivery of non-coding RNAs. Exosomes are microvesicles released by cells into the extracellular environment, measuring between 50 and 90 nm, and containing cholesterol, ceramide, sphingomyelin, and various proteins, including heat shock proteins such as HSP70 [192]. Microvesicles also contain thousands of heterogeneous multiple RNA species, including mRNA, microRNA, lncRNA, and circular RNA (circRNA). These RNAs can be delivered to different cells and be functional [193]; for example, miRNAs can serve as a guide by base-pairing with the target mRNA [194].

An interesting example of transfer is miR-142-3P; it modulates the innate immune response in innate immune effector cells of the central nervous system, known as microglia, through the Camk2a-Creb-BDNF pathway. Like other exogenous miRNAs, miR-142-3P is involved in cell proliferation, invasion, progression, metastasis, and drug resistance [195]. This is released from monocytes via exosomes when taken up by retinoblastoma (RB) cells; it inhibits the proliferation of the RB cell line [196,197].

Horizontal transfer of RNA can occur using platelet-derived microparticles (PMP) in lung and colon tumors [144,198]; for example, the transfer of miR-24, which is located in the mitochondria and inhibits the mitochondrial function of tumor cells via mitochondrial encoded NADH dehydrogenase 2 (mt-Nd2) and Small Nucleolar RNA, H/ACA Box 75) (Snora75) [163]. miR-24, miR-30b, and miR-142-3p regulate phagocytosis of myeloid inflammatory cells and enhance innate immunity [199,200].

On the other hand, miR-24-3p can be transferred from cancer-associated fibroblasts to colon cancer cells via exosomes and promotes the resistance of colon cancer cells to methotrexate [201].

Another instance of transferability in the development and spread of hepatocellular carcinoma (HCC) is seen in OncomiR (cancer-associated microRNA) derived from HCC extracellular vesicles that interact with hepatic stellate cells [202]. One more example is miR-223, which is highly expressed in hematopoietic cells, favoring granulopoiesis and at the same time reducing differentiation to macrophages; by negative feedback, miR-223 modulates the innate immune response [203]. miR-223 is transported by vesicles to monocytes, endothelial cells, epithelial cells, and fibroblasts [204]. In addition, it has been observed that both human macrophage miRNA-223 and miRNA-142 can be transferred to HCC cells where they inhibit HCC proliferation [205].

## 11. Interrelationship of Trained Immunity and ncRNA Could Modulate Their Effects on Cancer

The characteristics that favor the interrelation of trained immunity and ncRNA may explain how innate immunity can be regulated. It is known that ncRNAs are involved in the modulation of innate immunity such as “upstream master LncRNA of the inflammatory chemokine locus” UMLILO, a lncRNA that promotes the epigenetic priming of chemokines [206].

Two fundamental reprogrammings are involved in trained immunity, one epigenetic and the other metabolic [207]. These reprogrammings induce an increase in cytokines such as IL-6, TNF-α, IL-1β, IL-8, and monocyte chemoattractant protein-1 (MCP-1/CCL2); an increase in glycolysis, in fatty acid synthesis, and in epigenetic reprogramming and a decrease in Mono-methylation of lysine 4 on histone H3 (H3K4me1) and Trimethylation of Histone H3 at Lysine 4 (H3K4me3) is observed.

Among the signaling pathways of trained immunity, some of the most widely studied are ones induced by β-glucan through dectin-1/Akt/mTOR/HIF1α-dependent induction of aerobic glycolysis [208]. Another signaling pathway of the trained immunity is induced by BCG vaccine and muramyl dipeptide (MDP) through NOD/NFκβ [209,210]. JAK/STAT is also another pathway that is being studied in neutrophils [211].

As stated in Section 5**,** shown in Figure 2 is the case of the lncRNA CRNDE, miR-345-5p, in pre-leukemic B-cell clones. CRNDE has the opposite specificity, targeting miR-345-5p/cyclic AMP to promote cell growth in B cells (BCP-ALL). On the other hand, in chronic lymphocytic leukemia (CLL), CRNDE regulates the expression of NDRG2 through miR-28, where it suppresses the proliferation and stimulates apoptosis of MEG1 and HG3 cells [212].

A hypothetical case in which both systems could be involved is pancreatic cancer. In this cancer, miR-345-5p functions as a tumor suppressor by targeting C-C motif chemokine ligand 8 (CCL8) [213]; on the other hand, in the same cancer, the induction of NK cells by a D-glucan from *Strongylocentrotus nudus* eggs [214] can inhibit cancer growth by stimulating the TLR4/MAPKs/NF-κβ signaling pathway [215,216].

## 12. Strengths and Limitations

Our review is the first to integrate the features of specificity, inducibility, epimutation, transfer, inheritance, chronic inflammation of the innate immune response, and ncRNAs. Limitations of this work include the lack of review of other ncRNAs being studied in cancer such as small nuclear RNAs (snRNAs), small nucleolar RNAs (snoRNAs), ribosomal RNAs (rRNA), tRNA derived fragments (tRF), tRNA half (tiRNA) and telomerases (TERC), circRNAs and linear RNAs, and enhancer RNAs (eRNAs) [96].

## 13. Conclusions

Both defense mechanisms aim to eliminate foreign agents or aberrant genes such as transposons. However, errors in these systems can lead to various diseases, including cancer. The relationship between trained immunity, ncRNAs, inflammation, and cancer suggests that cancer treatment should include not only immunotherapy and anti-inflammatory control, but also ncRNA modulation. This comprehensive approach may lead to more effective cancer therapies. However, although the similarities between the innate immune response and ncRNAs are evident, further studies are needed to fully integrate these defense mechanisms and trained immunity into cancer treatment. In conclusion, the combination of ncRNA modulation with trained immunity and anti-inflammatory strategies holds great promise for advancing cancer treatment. Understanding the complex interactions between these systems and pathways is necessary to design and develop personalized targeted therapies.

## Figures and Tables

**Figure 1 biomedicines-12-02072-f001:**
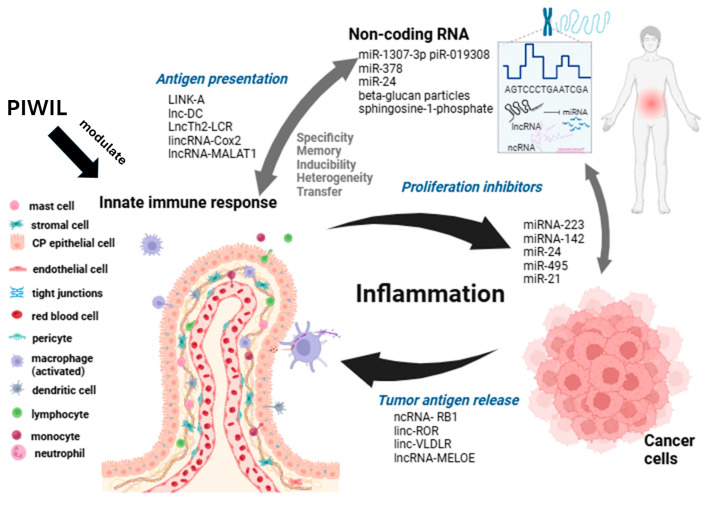
Dysregulation of siRNAs, lncRNAs, miRNAs, piRNAs, and innate immunity leads to chronic inflammation and cancer. PIWIL modulate cells such as neutrophils, monocytes, and dendritic and NK cells.

**Figure 2 biomedicines-12-02072-f002:**
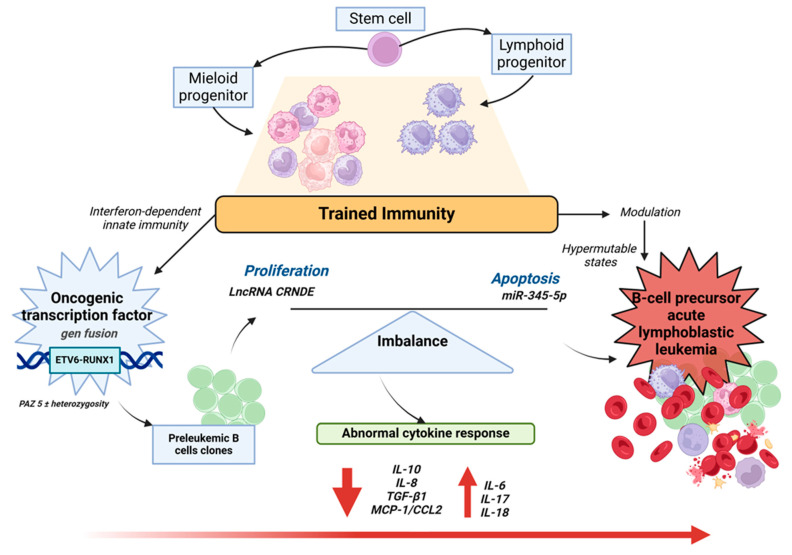
Specific regulation in the proliferation and apoptosis of pre-leukemic B-cell clones by lncRNA Colorectal neoplasia differentially expressed (CRNDE), miR-345-5p, and interferon-dependent trained immunity. Runt-related transcription factor 1 (RUNX1), transforming growth factor β1 (TGF-β1), monocyte chemoattractant protein-1 (MCP-1/CCL2).

**Table 1 biomedicines-12-02072-t001:** Main types of non-coding regulatory RNA (ncRNA) from the point of view of their structure and role in the regulation of innate immunity in cancer.

Type	Name	Size (nt Number)	Genetic Regulation Mechanisms	Example of Function in Innate Immune Response and Cancer	Examples of Potential Applications
lncRNA >200 nt	Linear long non-coding RNAs (Linear lncRNA)	>200	Activate or repress [32]	Overexpression of lncRNA H19 enhances carcinogenesis and metastasis of GC, GBC, PDAC, CRC, EC, OC, and NSCLC by regulating macrophage polarization from M1 to M2 phenotypes, and also regulates the immune cell activity and infiltration [33,34,35]	Plasma lncRNA H19 levels have been proposed as predictive biomarkers for cancers such as gastric, bladder, lung, and breast cancer [36,37]
	circular lncRNA (circRNA)	100–10,000	Sponge [38]	Innate immune cells in the tumor microenvironment can be modulated by aberrant expression of specific circRNAs such as circ_0000977 and circASAP1 [39,40]	circCAMSAP1 is a potential diagnostic and prognostic biomarker, as well as a possible therapeutic target in colorectal cancer [41]
	enhancer RNA (eRNA)	50–2000	Interact with transcriptional regulators [42]	Oncogenic super-enhancers (SEs) are involved in tumor metastasis. SEs could induce IL-20RA overexpression affecting cell proliferation and immune evasion-related gene expression [43,44]	Among the therapeutic agents associated with SEs that are currently in clinical trials are those related to hematologic malignancies and solid tumors [45]
mncRNAs 50 to 200 (or 400) nt	small nucleolar RNA (snoRNA)	50–300	pre-rRNA cleavage and 3′ mRNA splicing [46]	Regulate the overexpression of dendritic cells, macrophages, and induced cytokine to influence tumorigenesis and tumor immunity [47,48]	snoRNAs are potential diagnostic and prognostic markers for cancers such as chronic lymphocyte leukemia, colorectal cancer, and ovarian cancer [49]
	small Cajal body-specific RNA (scaRNA)	50–400	Splice [50]	scaRNA15 directs p53 and redox homeostasis via selective splicing in cancer cells, which is regulated after MYC hyperactivation [51]	scaRNA could also be used as a diagnostic marker and potential therapeutic target, such as SCARNA12 in bladder cancer [52]
sncRNA 20–50 nt	small interfering RNA (siRNA)	20–25	RNA interference, cleavage and degradation of mRNA [53]	Downregulate the expression of immunosuppressive immune checkpoints, either within the tumor environment or on immune cells [54]	The use of siRNA is effective in inhibiting the proliferation of cancer cells, for this reason, there are currently clinical trials using sRNA in the treatment of liver, colorectal, pancreatic, kidney, prostate, ovarian, skin and hematological cancers [55]
	microRNA (miRNA)	21–23	Repress and degrade [56,57]	Regulate the differentiation, activation, and effector functions of innate immunity cells that have important effects on cancer progression, such as miR-21, miR-138-5p, and miR-200a, b, and c [58,59]	Currently, several miRNA mimics and antisense miRNA inhibitors as well as other ncRNAs, such as miR-34a, miR-16, miR-155, miR-193a3p, and miR-10b, are in pre-clinical or clinical trials in cancer patients [60]
	piwi-interacting RNA (piRNA)	26–32	Slicer activity of Piwi domain [61]	Abnormal piRNA expression can influence the phosphoinositide 3-kinase (PI3K)/phosphatase and tensin homolog (PTEN)/protein kinase B (Akt)/mammalian target of rapamycin (mTOR) (PI3K/PTEN/Akt/mTOR), and Mitogen-activated protein kinase/ ERK kinase/extracellular-signal-regulated kinase (ERK) (Ras/Raf/MEK) pathways, which are crucial in gene regulation. Their stimulation can enhance the metabolism, growth, and survival of cancer cells [62,63,64]. Furthermore, piRNAs repress the major histocompatibility complex (MHC) class II that helps cancer cells avoid immune recognition and reaction, e.g., PIWIL4 in BC [65].	Among the different types of piRNAs associated with cancer prediction and prognosis are piR-54265 in colorectal cancer, piR-651 in lung cancer, piR-823 in kidney cancer, and piRNA-823 in colorectal cancer (CRC), renal cell carcinoma (RCC), and multiple myeloma (MM) [66]

Input data: Long non-coding RNA (lncRNA), Medium non-coding RNA or mid-size noncoding RNAs (mncRNAs), Small non-coding RNA (sncRNA), nucleotides (nt), gastric cancer (GC), gallbladder cancer (GBC), pancreatic ductal adenocarcinoma (PDAC), colorectal cancer (CRC), esophageal cancer (EC), ovarian cancer (OC), non-small-cell lung cancer (NSCLC), Super enhancers (SEs), breast cancer (BC), P-element induced wimpy testis like protein (PIWIL).

**Table 2 biomedicines-12-02072-t002:** Examples of some common features between the innate immune response and small non-coding RNAs in cancer.

Similarities	Innate Immunity	Small Non-Coding RNAs
Specificity	Using single-cell RNA sequencing (scRNA-seq) technologies, previous studies found that the regulation of Natural killer cells (NK cells) antitumor activity is mediated by a specific subpopulation of myeloid LAMP3+ dendritic cells [134].	In a study of circulating exosome small RNAs using unique a molecular identifier (UMI) small RNA sequencing, miR-1307-3p and piR-019308, in combination with carcinoembryonic antigen (CEA) and carbohydrate antigen (CA) are highly specific for gastric cancer [135].
Memory	It was reported by flow cytometry, RNA sequencing, and Single-cell ATAC (Assay for Transposase Accessible Chromatin) sequencing that β-glucan induction enhanced antitumor immunity in mice, accompanied by transcriptomic and epigenetic rewiring of granulopoiesis. Furthermore, neutrophils were also reprogrammed into an antitumor phenotype. Bone marrow transplantation transferred the antitumor effect to treatment-naïve recipient mice [136].	By comparing epigenetic profiles in male germ cells from five mammalian species and one avian species, it is reported that there is a parallel evolution of the epigenetic balance of the male germ line and somatic development in animals [137].
Inducibility	Utilizing flow cytometry, tumor metastasis models, RNA sequencing, RT-qPCR, transmission electron microscopy, and cytometry by Time-Of-Flight (CyTOF^®^), previous studies have found that trained immunity triggered by β-glucan particles is mediated by sphingosine-1-phosphate, resulting in anti-tumor activity and reduced metastasis in lung cancer mice [138].	Glucocorticoids (GCs) are administered to alleviate the side effects of chemotherapy; however, using RT-qPCR, transmission electron microscopy (TEM), immunohistochemistry, and in situ hybridization it has been shown that in some tumors, such as pancreatic ductal adenocarcinoma, the administration of GCs increases the progression and metastasis of this tumor. GCs induce epigenetic signaling through miR-378 [139].
Heterogeneity	Single-sample gene set enrichment analysis (ssGSEA) and bulk RNA-seq analyses are used to identify NK cell subtypes in the heterogeneous tumor immune microenvironment (TIME) of hepatocellular carcinoma [140].	The miRNA profile of patients with hepatocellular carcinoma is highly variable compared to that of healthy subjects [141].
Transfer	-----	Using tissue microarrays, immunohistochemistry, and nanoparticle tracking, previous studies showed that miR-24 transfer can modulate metastasis in some cancers, e.g., platelet-derived microparticles (PMPs) miR-24 inhibit the growth of ectopic lung and colon carcinoma tumors [142,143].

## Data Availability

Not applicable.

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
