# Peer review of "Non-Coding RNAs and Innate Immune Responses in Cancer"

_biomedicines, 2024, doi:10.3390/biomedicines12092072_

Round 1

Reviewer 1 Report

Comments and Suggestions for Authors

The article entitled “Non-coding RNAs and innate immune responses in cancer” by Carlos Romero Díaz et al. is a review of existing literature, summarising the highly interesting field of non-coding RNAs and their effect on the immune response in cancer. The current version of the manuscript is a bit weak and needs a significant improvement. The authors have to address some points before this study can be published in the journal “biomedicines”.

My concerns are:

- Very often citations for statements are missing; the authors have to prove all statements with citations. There are too many citations missing so that I cannot mention here all.

- For some important points, the authors provide only one citation and that is not completely accurate. They have to add much more citations of the important work that exist in this field.

- Sometimes more up to date citations must be used and more up to date data are available.

- When the authors write “There are numerous examples ….” or “Other cases of…..” they have to provide more than one citation as proof.

- The authors have to structure the chapter 2 much better. Furthermore, miRNA and the other scRNAs must be introduced here too.

- An abbreviation should only be introduced when it is also used later in the text. But if an abbreviation has been introduced, the authors have to use it in the following text. The authors have to check the manuscript in regard to these points. Furthermore, they have to summarise all abbreviations in alphabetical in an abbreviation list at the end of the text.

- The sentence “siRNAs have been proposed as tools, particularly for the treatment of tumors, due to their gene-silencing mechanisms” is true but it is a very general statement. Why is this added to a specific piRNA example?

- It is neither nice nor necessary to read “The work of Kalafati L et al. ….” etc. If somebody is interested in the name of the first author and year of the publication (s)he will find this information in the Reference list. The authors have to rephrase all these parts.

- The “Conclusion” part is very weak and must be rewritten.

Comments on the Quality of English Language

Minor editing of English language required

Author Response

Reviewer 1

The article entitled “Non-coding RNAs and innate immune responses in cancer” by Carlos Romero Díaz et al. is a review of existing literature, summarising the highly interesting field of non-coding RNAs and their effect on the immune response in cancer. The current version of the manuscript is a bit weak and needs a significant improvement. The authors have to address some points before this study can be published in the journal “biomedicines”.

Q1.  Very often citations for statements are missing; the authors have to prove all statements with citations. There are too many citations missing so that I cannot mention here all.

A1. We appreciate the comments, we have improved the manuscript and mention all the citations missing.

Q2.  For some important points, the authors provide only one citation and that is not completely accurate. They have to add much more citations of the important work that exist in this field.

A2. We have provided more citation and now they are completely accurate

Q3.  Sometimes more up to date citations must be used and more up to date data are available.

A3. We have updated appointments and data. to ensure that our work reflects the current state of knowledge and accurately represents any recent developments or changes in the field.

Q4.   When the authors write “There are numerous examples ….” or “Other cases of…..” they have to provide more than one citation as proof.

A4. We thank the reviewer for this comment. We have provided more than one citation.

Q5.  The authors have to structure the chapter 2 much better. Furthermore, miRNA and the other scRNAs must be introduced here too.

A5. To address this point raised by the reviewer, we have now structured chapter 2 and included an introduction of miRNAs and other cRNAs.

Q6.  An abbreviation should only be introduced when it is also used later in the text. But if an abbreviation has been introduced, the authors have to use it in the following text. The authors have to check the manuscript in regard to these points. Furthermore, they have to summarise all abbreviations in alphabetical in an abbreviation list at the end of the text.

A6. We thank the reviewer for this comment. We have made the requested corrections.

Q7.   The sentence “siRNAs have been proposed as tools, particularly for the treatment of tumors, due to their gene-silencing mechanisms” is true but it is a very general statement. Why is this added to a specific piRNA example?

A7. We have rearrenged most of the paragraphs so that the information is in accordance with the section title.

Q8.  It is neither nice nor necessary to read “The work of Kalafati L et al. ….” etc. If somebody is interested in the name of the first author and year of the publication (s)he will find this information in the Reference list. The authors have to rephrase all these parts.

A8.  We have restructured all the sentences.    

Q9.  The “Conclusion” part is very weak and must be rewritten.

A9.  We have rewritten the conclusion

Reviewer 2 Report

Comments and Suggestions for Authors

In this study, Carlos et al. describe the relationship of non-coding RNAs and innate immune responses in cancer. They have investigate the properties of inducibility, specificity, memory between them. They also describe the classification and function of the non-coding RNAs. This review work is relatively acceptable to publish in our journal. However, some revisions were needed to make the manuscript better.

Main comments:

1. There are many types of cancers, and the cancers are distinctly different from each other. The authors need to specially focus on some types of cancers, which make the review more reliable.

2. Please add a paragraph to describe the different mechanism of non-coding RNAs in cancers

3. Please describe why non-coding RNA have similar process with innate immune response in cancer?

Author Response

Reviewer 2

In this study, Carlos et al. describe the relationship of non-coding RNAs and innate immune responses in cancer. They have investigate the properties of inducibility, specificity, memory between them. They also describe the classification and function of the non-coding RNAs. This review work is relatively acceptable to publish in our journal. However, some revisions were needed to make the manuscript better.

 Q1.   There are many types of cancers, and the cancers are distinctly different from each other. The authors need to specially focus on some types of cancers, which make the review more reliable.

A1.  We agree with the reviewer, we have focused on a type of Cancer

Q2.   Please add a paragraph to describe the different mechanism of non-coding RNAs in cancers

A2. We have added a paragraph describing the different mechanism of non-coding RNAs in cancers

Q3.   Please describe why non-coding RNA have similar process with innate immune response in cancer?

A3.  We have added a paragraph describing why non-coding RNA have similar process with innate immune response in cancer.

Reviewer 3 Report

Comments and Suggestions for Authors

Overview
NcRNAs play crucial roles in regulating gene expression and cellular processes, and they have
emerged as significant players in the context of cancer and innate immune responses. Several
research articles and reviews have made significant contributions. The present review aims to
highlight the role of different types of ncRNAs in innate immune response associated with
cancer.
Major Comments
1. Abstract should be re-written. Abstract provides the context for the need of the review,
but does not offer any information regarding what the present review is offering. Should
clearly highlight the topics covered in the review.
2. Introduction: Should discuss the last two paragraphs of the section under separate
heading “Inlcusion and exclusion criteria.”
3. Table 1: Column 2 and 3 is repetitive in Line 82 to 88. Can be removed. Column 4 is
just size and Column 5 “Genetic regulation mechanisms” provides basic information.
Overall, Table 1 does not offer significant information.
4. Most of the information is basic from Section 3 to Section 12. No constructive
discussion based on the relevant research articles was observed.
5. Should include more tables highlighting different ncRNAs and their importance
reported in innate immune response in various cancers.
6. Conclusion is not satisfactory. Should highlight the limitations and future prospects as
well.
Minor Comments
1. Line 80, 92: ncRNAs, already abbreviated.
2. Line 88: lncRNA, already abbreviated in Line 82.
3. Line 125: siRNAs, already abbreviated.
4. Figure 1 is blurry, better quality should be provided.
5. Line 278: It should be Table 2.
6. Line 301: Reference error.
Remark
Review provides the context for the need of the review and basic knowledge, but does not
provide satisfactory discussion of the relevant research outcomes reported in existing literature.

Author Response

Reviewer 3

NcRNAs play crucial roles in regulating gene expression and cellular processes, and they have emerged as significant players in the context of cancer and innate immune responses. Several research articles and reviews have made significant contributions. The present review aims to highlight the role of different types of ncRNAs in innate immune response associated with cancer.

Q1.   Abstract should be re-written. Abstract provides the context for the need of the review, but does not offer any information regarding what the present review is offering. Should clearly highlight the topics covered in the review.

A1. We thank the reviewer for this comment. We have made the requested corrections.

Q2.   Introduction: Should discuss the last two paragraphs of the section under separate heading “Inlcusion and exclusion criteria.”

A2. We have rewritten the introduction. We now add inclusion and exclusion criteria.

Q3.   Table 1: Column 2 and 3 is repetitive in Line 82 to 88. Can be removed. Column 4 is just size and Column 5 “Genetic regulation mechanisms” provides basic information. Overall, Table 1 does not offer significant information.

A3.  We have restructured table 1

Q4.   Most of the information is basic from Section 3 to Section 12. No constructive
discussion based on the relevant research articles was observed.

A4.  We have restructured all the sentences in section 3 to 12.

Q5.   Should include more tables highlighting different ncRNAs and their importance
reported in innate immune response in various cancers.

A5. We have restructured and complemented the information in table 1

Q6.   Conclusion is not satisfactory. Should highlight the limitations and future prospects as well.

A6.  We have rewritten the conclusion

Minor Comments

  1. Line 80, 92: ncRNAs, already abbreviated.
    2. Line 88: lncRNA, already abbreviated in Line 82.
    3. Line 125: siRNAs, already abbreviated.
    4. Figure 1 is blurry, better quality should be provided.
    5. Line 278: It should be Table 2.
    6. Line 301: Reference error.

We agree with the reviewers, we have corrected these minor comments.

Remark
Review provides the context for the need of the review and basic knowledge, but does not provide satisfactory discussion of the relevant research outcomes reported in existing literature.

We appreciate the comments. We have improved the manuscript.

Round 2

Reviewer 1 Report

Comments and Suggestions for Authors

The article entitled “Non-coding RNAs and innate immune responses in cancer” by Carlos Romero Díaz et al. is a review of existing literature, summarising the highly interesting field of non-coding RNAs and their effect on the immune response in cancer. The manuscript has been improved in the first round of revision but is still not good enough for publication. The authors have addressed only some of my concerns and they even have done this only partially. Broad parts of the manuscript must still be modified before this study can be published in the journal “biomedicines”. I don`t like to repeat myself and ask the authors to refer to my previous concerns – all these points have to be addressed end-to-end in the whole manuscript. This hasn`t done up to now and therefore this manuscript cannot be published. A second round of revision is a kind offer and if the issues are not solved after the second revision the manuscript should be rejected.

My further concerns with the modified version are:

- The abstract is hard to read due to the complex and extremely long sentences. It must be improved.

- Abbreviation must be introduced when they are used for the first time (one example is lines 76 and 82) - Please note it is only one example of several failures.

- The authors have to unify the style how they introduce an abbreviation. Either first the abbreviation followed by the full name in brackets or the full name followed by the abbreviation in brackets. It is not acceptable to mix these styles.

- The statement “…..and treat many types of cancer” in Table 1 is not acceptable in this context.

- The authors have to explain why they repeat the same sentence in lines 179-186.

- The authors have to justify why they use these four examples in lines 260-295. In my opinion there are much better examples which could be used here.

- The authors appear to be a bit lazy and haven`t proof read the manuscript before re-submission as underlined by “[135Error! Bookmark not defined]”. This is something you should notice before submission of a manuscript.

- The “Conclusions” have been slightly improved but must be further improved.

Comments on the Quality of English Language

 Minor editing of English language required.

Author Response

Reviewer 1

The article entitled “Non-coding RNAs and innate immune responses in cancer” by Carlos Romero Díaz et al. is a review of existing literature, summarising the highly interesting field of non-coding RNAs and their effect on the immune response in cancer. The manuscript has been improved in the first round of revision but is still not good enough for publication. The authors have addressed only some of my concerns and they even have done this only partially. Broad parts of the manuscript must still be modified before this study can be published in the journal “biomedicines”. I don`t like to repeat myself and ask the authors to refer to my previous concerns – all these points have to be addressed end-to-end in the whole manuscript. This hasn`t done up to now and therefore this manuscript cannot be published. A second round of revision is a kind offer and if the issues are not solved after the second revision the manuscript should be rejected.

My further concerns with the modified version are:

The abstract is hard to read due to the complex and extremely long sentences. It must be improved.

Answer: We have improved the style of the text and shortened the sentences in the abstract, thanks again to the reviewer.

Abbreviation must be introduced when they are used for the first time (one example is lines 76 and 82) - Please note it is only one example of several failures.

Answer: We thank the reviewer for this comment. We have made the requested corrections.

The authors have to unify the style how they introduce an abbreviation. Either first the abbreviation followed by the full name in brackets or the full name followed by the abbreviation in brackets. It is not acceptable to mix these styles.

Answer: Thank you for your detailed comments about the acronyms and their style. They have all been corrected.

 The statement “…..and treat many types of cancer” in Table 1 is not acceptable in this context.

Answer: Thanks for pointing this out. Table 1 has been fixed and improved.

The authors have to explain why they repeat the same sentence in lines 179-186.

Answer: We are sorry, but this error has now been corrected.

The authors have to justify why they use these four examples in lines 260-295. In my opinion there are much better examples which could be used here.

Answer: You have raised an important point here. At the reviewer's suggestion, we changed the examples.

The authors appear to be a bit lazy and haven`t proof read the manuscript before re-submission as underlined by “[135Error! Bookmark not defined]”. This is something you should notice before submission of a manuscript.

Answer: We thank the reviewer for this comment. Sorry for not catching "Error! Bookmark not defined" Thanks

- The “Conclusion” part is very weak and must be rewritten.

Answer: Thank you for pointing this out. I agree with this comment. The conclusion has now been rewritten and improved.

Reviewer 2 Report

Comments and Suggestions for Authors

The authors have well replied my comments, thus I recommend accept in this version.

Author Response

(The authors gave the same response as above.)

Round 3

Reviewer 1 Report

Comments and Suggestions for Authors

The authors have tried to improve the manuscript but they failed to address all the concerns even in the second round of revision. Once again they appear to be a bit lazy and haven`t proof read the manuscript before re-submission as underlined by two “[Error! Bookmark not defined]”. This is something you should notice before submission of a manuscript.

Furthermore, they still introduce one and the same abbreviation twice and they haven`t unify the style how they introduce an abbreviation (one example is in table2).

Therefore, this manuscript should be rejected

Comments on the Quality of English Language

Minor editing of English language required.